# Protocol for a feasibility randomised controlled trial of a musculoskeletal exercise intervention versus usual care for children with haemophilia

Ferhana Hashem,[1] Melanie Bladen,[2] Liz Carroll,[3] Charlene Dodd,[4]
Wendy I Drechsler,[5] David Lowery,[1] Vishal Patel,[6] T Pellatt-Higgins,[1] Eirini Saloniki,[1]
David Stephensen [4,6]

¹Centre for Health Service Studies, University of Kent, Canterbury, Kent, UK
²Haemophilia Centre, Great Ormond Street Hospital For Children NHS Foundation Trust, London, UK
³Haemophilia Society, London, UK
⁴Haemophilia Centre, East Kent Hospitals University NHS Foundation Trust, Canterbury, UK
⁵School of Population Health and Environmental Sciences, King's College London, London, UK
⁶Haemophilia Centre, Barts Health NHS Trust, London, UK

**Correspondence to**
Dr David Stephensen;
david.stephensen@nhs.net

## ABSTRACT

**Introduction** Haemophilia is a rare, inherited disorder in which blood does not clot normally, resulting in bleeding into joints and muscles. Long-term consequence is disabling joint pain, stiffness, muscle weakness, atrophy and reduced mobility. The purpose of this proposed feasibility of a randomised controlled trial (RCT) is to test the feasibility of an age-appropriate physiotherapy intervention designed to improve muscle strength, posture and the way boys use their joints during walking and everyday activities.

**Methods and analysis** A small-scale two-centre RCT of a 12-week muscle strengthening exercise intervention versus usual care for young children with haemophilia will be conducted. Primary outcomes will be safety and adherence to the exercise intervention. Secondary outcomes will include recruitment, retention and adverse event rates, clinical data, muscle strength, joint biomechanics and foot loading patterns during walking, 6 min timed walk, timed-up-and-down-stairs, EQ-5D-Y, participants' perceptions of the study, training requirements and relevant costs. Recruitment, follow-up, safety and adherence rates will be described as percentages. Participant diary and interview data will be analysed using a framework analysis. Demographic and disease variable distributions will be analysed for descriptive purposes and covariant analysis. Estimates of differences between treatment arms (adjusted for baseline) and 75% and 95% CIs will be calculated.

**Ethics and dissemination** The study has ethical approval from the London—Fulham Research Ethics Committee (17/LO/2043) as well as Health Research Authority approval. As well as informing the design of the definitive trial, results of this study will be presented at local, national and international physiotherapy and haemophilia meetings as well as manuscripts submitted to peer-reviewed journals. We will also share the main findings of the study to all participants and the Haemophilia Society.

## Strengths and limitations of this study

► This study investigates the feasibility of a multicentre randomised controlled trial of a muscle strengthening exercise intervention versus usual care in children with haemophilia.
► The intervention to be studied has been coproduced by children with haemophilia, their parents and healthcare professionals.
► It is impossible to blind those in receipt and those delivering the intervention as to the group allocation, but we will blind the assessor as to group allocation.
► At the end of this study, we will not be able to answer the question 'Does exercises help improve the long term health of children with haemophilia?', but we will know whether such a study could be undertaken and the best way to deliver the intervention.

bleeding into joints and muscles is the result; the ankle joint being the most common.[1] Long-term consequence is chronic arthropathy, a pattern of disabling joint pain, stiffness, muscle weakness, atrophy and reduced mobility.[2–4] The introduction of prophylaxis with clotting factor concentrates has reduced the number of bleeding episodes experienced by children, changing their lifestyle and long-term outlook.[5 6] However, evidence indicates that despite prophylaxis, boys still bleed one to two times per year, with relatively few bleeding episodes being required to initiate bone destruction, and in some cases, articular erosion is evident with no history of joint bleeding.[1 7–9] Furthermore, children are more susceptible to blood-induced damage than adults and once established is progressive and irreversible.[10]

It is well established that muscle weakness is associated with haemophilic arthropathy in adults,[4] and it is now becoming apparent that muscle strength is reduced in children,

## INTRODUCTION

Haemophilia is a rare, inherited disorder in which blood does not clot normally. Repeated spontaneous and traumatic

prior to the onset of clinical arthropathy.[11] In a study of 10 adolescent and adult haemophiliacs with a history of unilateral knee bleeding, Falk *et al* found that strength of the knee extensors and flexors in haemophilic children was significantly weaker than those of the age-matched controls.[12 13] More recently, Stephensen *et al* observed that despite normal clinical examinations and comparable levels of physical activity, young haemophilic boys with a history of ankle joint haemorrhage had significantly reduced dynamic muscle strength together with atrophy of the knee extensors, ankle dorsiflexors and plantarflexors when compared with age-matched and size-matched typically developing peers.[11] Furthermore, muscle weakness of the leg is strongly correlated to muscle atrophy, reduced walking distances, slower ascent and descent of stairs, as well as reduced ankle joint motion, greater knee flexion motion and forces at the knee and ankle joint during walking.[14 15]

Therapeutic exercise is an important component of the management of other forms of arthropathy (eg, osteoarthritis, rheumatoid arthritis),[16] and it would appear logical that exercise would be effective for people with haemophilia. There is growing evidence of the benefits of incorporating muscle strengthening exercise in their management of arthropathy.[17] It is the view of clinicians that increases in muscular strength and postural control might improve motor performance and cardiovascular fitness, limit exaggerated end-range joint movement and promote optimal transfer of weight-bearing forces through joints, thereby minimising muscle imbalance, synovial impingement and associated haemarthroses or synovitis.[18] However, there is a lack of evidence to support these assumptions. A recent Cochrane Review evaluating the safety and effectiveness of exercise for people with haemophilia reported four randomly controlled studies of an exercise intervention in children with the condition and concluded the studies were of low or very low quality, due to small sample sizes and potential bias.[17] Furthermore, no paediatric study compared a muscle strengthening intervention to a control group or intervention without muscle strengthening exercises. Several studies compared two multicomponent exercise interventions that included broad muscle strengthening as well as treadmill walking and cycle ergometry, and it is not possible to determine the effect of muscle strengthening on muscle strength or its effect on physical function.[19 20] Furthermore, preadolescent and adolescent boys aged between 10 and 14 years were included in these studies, and it is not known if the groups were matched for pubertal status.

Despite the apparent benefit, there is a lack of robust evidence to determine whether muscle strengthening exercise can improve or negatively affect outcomes for young children with haemophilia. The purpose of this proposed feasibility RCT is to test the feasibility of an age-appropriate physiotherapy intervention designed to improve muscle strength, posture and the way boys use their joints during walking and everyday activities.

## METHODS AND ANALYSIS

The protocol version for this feasibility study is V.2—19 December 2017. The Standard Protocol Items: Recommendations for Interventional Trials (SPIRIT) guidelines[21] have been used to report the study protocol.

### Participants

Participants will be invited from two haemophilia centres. Twenty boys with severe or moderate haemophilia A or B with or without inhibitors on prophylactic treatment with coagulation factor concentrate, aged 6–11 years, will be recruited. Boys with or without symptoms of joint damage (evaluated using the Haemophilia Joint Health Score and Haemophilia Early Arthropathy Detection with Ultrasound score) will be eligible to participate. Exclusion criteria are as follows: von Willebrand disease, history of fracture or trauma to the lower limb, orthopaedic surgery, acquired brain injury or any other disturbance of the central nervous system, joint or muscle bleed in the lower limb in the past 6 weeks, presence of lower limb pain or unable to fully comply with verbal instructions.

Participants will be identified from each Haemophilia Centre's database of registered patients. The parents or guardians of eligible participants will be approached initially by the principle investigator at each site via telephone to provide a brief verbal explanation of the research study and ask if they would be happy to receive a patient information sheet. Should they be interested in participating in the study they will receive, by post, the patient information sheet (one for parents and a separate age-appropriate version for children) at least 1 week prior to their next clinic appointment. This will allow for a 'cooling-off' period of a minimum of 1 week for the potential participants. The researcher will then be available for questions by telephone or at the participant's next clinic appointment.

At their clinic appointment, the investigator will meet with the participant and parent/legal guardian to complete the consent form as appropriate. The investigator will further explain the study procedures, why they are being asked to consider including their child in this study and the associated risks and benefits of being included. They will then be given an opportunity to ask any further questions. The child/parent/legal guardian will then be asked to confirm if the potential participant agrees to take part in the study. If consent is agreed, a parental consent form will be completed by the parent/guardian and a copy will be provided. If at any point during the study process the parent/guardian does not consent to inclusion in the study, then any data collected will be destroyed. Any decision to refuse or withdraw from the study will in no way influence the services that they receive. Once the child's parent/guardian has given consent, the child must also indicate that they do not object to participating in the research activity. This participant 'assent' will be sought.

## Study design

Following written informed consent and screening of eligibility criteria, randomisation of eligible participants for the study will be performed with the online randomisation service http://sealedenvelope.com/. Utilising the secure, centralised and independent service, participants will be randomly allocated into one of the two groups (treatment group 1 will receive a 12-week muscle strengthening exercise intervention; treatment group 2 will receive usual physiotherapy care for 12 weeks) on a ratio of 1:1. Allocation will be stratified by centre to ensure the same numbers of participants at each site are allocated to the two groups. The unblinded principal investigator at each site will be issued with a password to access the service to create a new randomisation for the study participants. Those allocated to treatment group 2 will be offered the intervention after their end of study follow-up assessment. Data will be collected prior to and immediately postintervention (figure 1).

It is impossible to blind those in receipt and those delivering the intervention as to the group allocation. However, it will be possible—although perhaps challenging—to blind the assessor as to group allocation. Consequently, we will attempt a single blind approach in which the participant and physiotherapists delivering the intervention will be encouraged to withhold their group allocation from the assessors collecting data. We will use a range of strategies, including instruction in the participant literature, intervention protocol and with frequent verbal prompts. In order to assess the efficacy of blinding, we will ask assessors to record which group they think participants have been allocated to at the end of each assessment. This will be presented as a binary forced choice. There are no circumstances under which it would be necessary to unblind assessors.

## Intervention

The intervention developed by expert clinicians and patients utilising a modified Nominal Group Technique (NGT) is a 24-session, 12-week programme designed as 2-week progressive levels (intensity and or load) with no more than 10 exercises in each session, taking 25–30 min to complete (figure 2). The intervention aims to master movement control and emphasises body-weight strength development. Pictures of the exercises and instructions for each phase will be provided in 2-week exercise diaries. Participants will be asked to complete the exercises twice per week; once with the physiotherapist who will visit the participant at home and once supervised by their parents/guardian. Completion of exercises, along with treatment regimen, any adverse events and comments in relation to the exercise programme will be recorded in the exercise diary. They will also record participation in physical activities, including type, time and intensity. According to predetermined criteria (safe and competent), the physiotherapist will determine if it is appropriate for the participant to progress to the next exercise level. On return

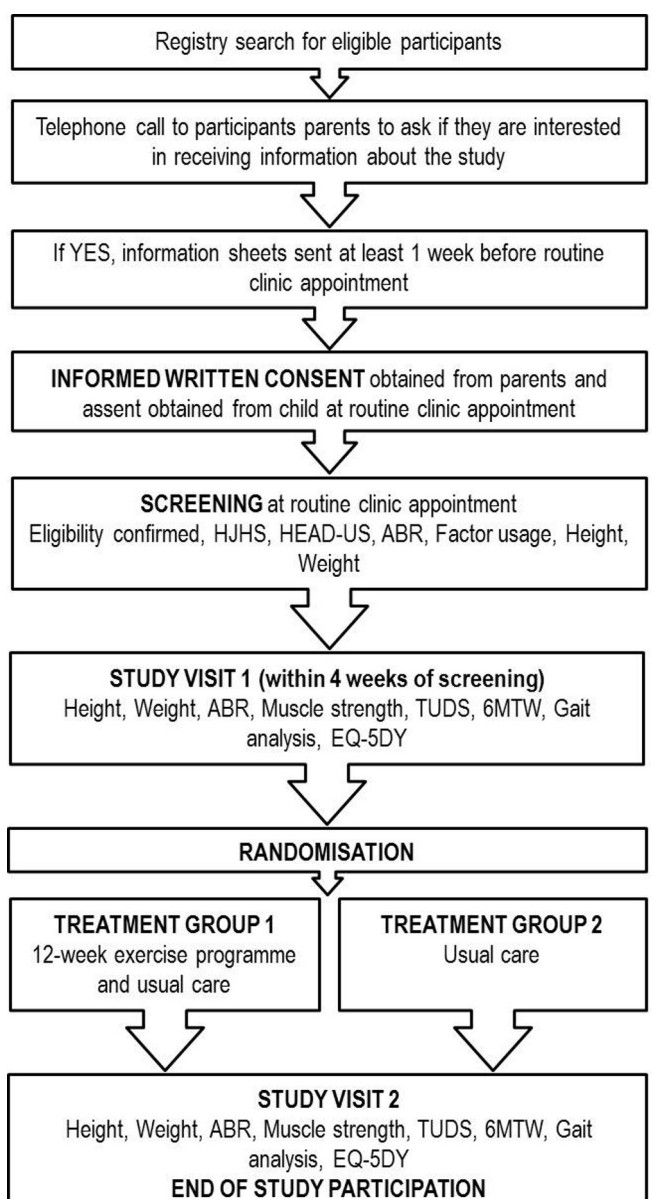

**Figure 1** Flowchart of participant journey through the feasibility study. ABR, annualised bleeding rate; HEAD-US, Haemophilia Early Arthropathy Detection with Ultrasound; HJHS, Haemophilia Joint Health Score; TUDS, timed-up-and-down-stairs; 6MWT, 6-minute timed walk.

of the exercise diary every 2 weeks, participants will be provided with a gift voucher.

## Usual care

Participants will continue to receive their usual care as recommended and prescribed by the haematologist, physiotherapist and other members of the healthcare team. This will include factor concentrate treatment, any physiotherapy or surgical intervention or advice that is recommended by the clinical care team. There are currently no standardised treatment recommendations for children without apparent joint damage. The researchers will not make or initiate any changes to routine treatment. Any change to or initiation of new treatment or intervention

| | Stretching | Balance | Strengthening | Functional activities |
|---|---|---|---|---|
| **Week 1-2** | Triangle Stretch<br>Octopus Forward Stretch<br>Turtle Stretch | Tree Pose<br>Football Pose | Assisted double-leg heel raises (2 sets of 5)<br>Foot caterpillar<br>Clamshell<br>Bridge – double leg | Heel toe walk |
| **Week 3-4** | Triangle Stretch<br>Octopus Forward Stretch<br>Turtle Stretch | Tree pose<br>Football pose (playing hand keepy-up with a balloon) | Double-leg heel raises (2 sets of 5)<br>Foot caterpillar<br>Clamshell<br>Bridge – single leg | Heel toe walk<br>Toe heel walk |
| **Week 5-6** | Triangle Stretch<br>Octopus Forward Stretch<br>Turtle Stretch | Tree pose on cushion<br>Football pose on cushion | Assisted single leg heel raises (2 sets of 5)<br>Three-legged dog<br>Crab Pose | Toe heel walk<br>Step ups |
| **Week 7-8** | Triangle Stretch<br>Octopus Forward Stretch<br>Turtle Stretch | Tree pose on cushion<br>Football pose on cushion (playing hand keepy-up with a balloon) | Single-leg heel raises (2 sets of 5)<br>Three-legged dog<br>Forward Warrior Pose | Step ups<br>Crab walk |
| **Week 9-10** | Triangle Stretch<br>Octopus Forward Stretch<br>Turtle Stretch | Tree pose with eyes closed<br>Scuba diver | Single-leg heel raises (3 sets of 5)<br>Two-legged Dog<br>Forward warrior pose<br>Sideway Warrior Pose | Crab walk<br>Side step ups |
| **Week 11-12** | Triangle Stretch<br>Octopus Forward Stretch<br>Turtle Stretch | Tree pose on a cushion with eyes closed<br>Scuba diver on cushion | Single-leg heel raises (3 sets of 5)<br>Forward Warrior Pose<br>Sideway Warrior Pose<br>Crab Pose | Step ups<br>Side step ups |

**Figure 2** Description of the exercise intervention.

will be recorded and the treating clinician will determine whether it is appropriate for the participant to continue in the study. Participants will be provided with a diary to record participation in physical activities, including type, time and intensity. Participants will be telephoned weekly to encourage dairy completion. On return of the diary every 2 weeks, participants will be provided with a gift voucher.

### Outcome measures
Primary outcome measures are as follows:
► Safety will be evaluated using annualised bleeding rates, coagulation factor usage (what the patient actually administers and records on their routine electronic diary) and reported adverse events and reactions associated with the intervention. Association with symptoms of joint disease (Haemophilia Joint Health Score and Haemophilia Early Arthropathy Detection with Ultrasound score) will be evaluated.
► Adherence to the exercise intervention will be evaluated by the number of exercise session completed as a percentage of the prescribed exercise sessions. Data will be extracted from participant diaries.
  Secondary outcome measures are as follows:
► Recruitment rate determined as the participants who volunteer to participate in the study as a percentage of those that are invited/eligible.
► Willingness to be randomised; evaluated at the end of study through semistructured interviews that will be audio-recorded and be transcribed by a member of the research team.

► Participant's acceptability of the trial and intervention: using semistructured interviews, conducted at the completion of the study, the participants will be asked their views on acceptability of the trial and intervention, recruitment and exercise adherence. The interview will be audio-recorded and transcribed by a member of the research team.
► Training required to deliver the intervention and collect outcome data at study visits: using semistructured interviews, conducted at the completion of the study, the physiotherapists will be asked their views on acceptability of the trial and intervention, recruitment and exercise adherence. They will also be asked about the training requirements for delivering the intervention as well as the appropriateness of the outcomes piloted in terms of training, length of time to complete and clinical value of the data. The interview will be audio-recorded and transcribed by a member of the research team.
► Haemophilia Joint Health Score (HJHS)—a clinical examination of swelling, muscle atrophy, strength, joint crepitus, range of motion and pain at the ankle, knee and elbow joints. The HJHS is a standardised multi-item assessment, currently used in clinic as part of routine care to record musculoskeletal health.[22]
► Haemophilic synovitis and arthropathy—the presence and severity of signs of haemophilic synovitis and arthropathy will be evaluated using the Haemophilia Early Arthropathy Detection with Ultrasound (HEAD-US) score. The HEAD-US protocol is a standardised

scanning protocol used to identify the presence and severity of synovitis, articular cartilage and chondral bone defects in the ankle, knee and elbow joint. The scanning procedure takes approximately 2–3 min per joint and we have recently shown good inter-rater repeatability of this protocol when performed by physiotherapists.[23 24]

► Muscle strength of the knee extensors and ankle dorsiflexors and plantarflexors—maximum torque will be recorded with a hand-held dynamometer (Manual Muscle Tester, Lafayette Instrument, USA) using our standardised protocol with excellent test-re-test repeatability.[15]

► Joint biomechanics and motion during walking—temporal spatial and three-dimensional (3D) joint motion during gait will be recorded using an inertial magnetic measurement gait analysis system (MVN BIOMECH Awinda, Xsens Technologies, Netherlands). Sensor units will be attached to the feet, lower legs, thighs and pelvis with elastic straps. Motion of the sensors will be used to calculate temporal spatial and 3D lower limb joint motion.

► Static and dynamic foot loading patterns—will be evaluated with a plantar pressure platform (HR Mat, Tekscan, USA). Centre of pressure motion in respect of distance and direction travelled, together with contact time as a measure of foot symmetry of foot loading for various foot segments (heel, midfoot and forefoot) along with distribution of peak pressure and pressure-time integrals will be evaluated.

► Six min timed walk—distance walked in 6 min. Two marker cones will be placed 30 m apart along a corridor and each subject will be asked to walk back and forth between the cones as fast as they can for 6 min.[15]

► Timed-up-and-down-stairs—time taken to ascend and descend a flight of 12 steps. The children will stand 30 cm from the bottom step and asked to walk quickly but safely up the stairs, turn around on the top step and come all the way down until both feet land on the bottom step. They will be instructed to step on every step, use the hand rail if they want and not to run.[15]

► EQ-5D-Y—a five dimension (mobility, self-care, usual activities, pain/discomfort, anxiety/depression) questionnaire measuring health-related quality of life.[25]

► Costs relevant to the study—intervention costs, health and social care costs, out-of-pocket expenses and time off work costs. Healthcare costs (using National Health Service reference costs were possible) will include training and delivering the intervention, equipment, overhead costs, visits to hospital/haemophilia clinic, general practitioner, other healthcare services, investigations, hospitalisation, treatment, coagulation clotting factor concentrate and medication. Patient and family costs will include travel and time off work. Feasibility of collecting these data will be explored along with the resource consequences.

Data will be collected at the respective haemophilia centres. All data collected related to the study will be confidential and anonymous. The participant will be allocated a randomised participant number code, and all data collected will relate to this code. A record of which code belongs to which subject will be stored in a locked cabinet separate from the study data. Only the principle investigator (who will be a member of the participant's direct care team) will have access to this documentation. The direct care team and the principal investigator will have access to participant's personal and identifiable information during the study. Consent will be sought by the lead researcher for the research team to have access to this personal information.

### Analysis methods

Recruitment, follow-up, safety and adherence rates will be described as percentages. Participant diary and interview data will be analysed to identify factors that influence these rates and acceptability of the trial using a framework analysis. Interview data from physiotherapists will be analysed using a framework analysis to determine training required to deliver the intervention and collect outcome data. Feasibility of collecting healthcare cost data will be explored along with the resource consequences. Costs will be analysed and reported using descriptive statistics. Demographic and disease variable distributions will be analysed for descriptive purposes and covariant analysis. Walking variables will be normalised to leg length and muscle strength to body mass. Group mean and SD (median and interquartile range where appropriate) will be calculated and reported. Change from baseline data will be compared between groups at all study time points using analysis of covariance with baseline as a covariate and a factor for centre. Estimates of differences between treatment arms in the changes from baseline (adjusted for baseline) and 75% and 95% CIs will be calculated. Group mean and SD (median and interquartile range where appropriate) will be calculated preintervention and postintervention. These data will allow us to estimate the potential effect of the intervention and provide estimates of variability to inform the sample size required for a full RCT, as well as to refine the outcome measures by evaluating their responsiveness to changes in musculoskeletal health.

### Patient and public involvement

Ten specialist haemophilia physiotherapists in the UK plus those experienced in sports injury were invited to take part in an NGT consensus discussion to share their experience and expertise to contribute to the development of the exercise intervention. Patients' priorities, experience and preferences informed the study design, recruitment as well as the design and delivery of the intervention. The exercise programme was demonstrated to five boys with haemophilia and their parents who were members of the Haemophilia Society, National UK Patient Organisation. The parents and children were asked about what

they thought about the exercises and whether they could undertake them on a regular basis, where they thought the best place was for undertaking them and how they would like to receive the information on the exercise programme. Parents and children were then asked questions about how they would feel about taking part in study testing the benefits of the exercises, issues around being allocated randomly into study groups and then questions about what would encourage the children to continue on the exercise programme with discussions focusing on types of rewards and incentivisation.

## Management and oversight of the study

East Kent Hospital University Foundation Trust is the study sponsor. They have overall responsibility for governance of the study and have no direct role in study design, data collection, management, analysis, interpretation of data and reports and publications that arise from the study.

A Research Monitoring Committee (RMC) has been established and will meet 6 monthly after the study has commenced. The RMC will be responsible for monitoring milestones and targets, data, safety, recruitment, reviewing and interpreting the results. The RMC will be chaired by an independent chair from outside the research team group. Additional meetings may be called to review any serious adverse events, adverse events (AEs) and adverse reactions (ARs). The chief investigator will co-ordinate day to day running of the research, complete ethical approvals, communicate with recruitment sites, complete training of physiotherapists to deliver the intervention, undertake research initiation and monitoring site visits, co-ordinate data analysis and interpretation. The chief investigator will notify the RMC in the case of AEs and ARs. The principal investigator at each site will be the physiotherapist delivering the intervention. They will be responsible for recruitment and screening participants according to the inclusion and exclusion criteria, reporting AEs and ARs and local governance requirements.

## ETHICS AND DISSEMINATION

Study newsletters will be published by the RMC and available on the website of the Haemophilia Society. The results of this feasibility study will inform the design of a definitive RCT to test whether muscle strengthening can benefit musculoskeletal health in children with haemophilia, including whether such a study can be achieved and the best way of completing it. The results of the feasibility study will be presented at local, national and international physiotherapy and haemophilia meetings as well as manuscripts submitted to peer-reviewed journals. The main findings of the study will be shared to all participants and the Haemophilia Society.

**Acknowledgements** The authors would like to acknowledge the contribution of the academic experts and specialist physiotherapists (Wendy Drechsler, Melanie Bladen, Danny Armitage, Trupti Bhandari, Lisa Geuran, Hannah Harbidge, Nicola Hubert, Ann McCarthy, Julie Sparrow, Linda Walsh), the children and parents of the Haemophilia Society for sharing their priorities, experience and preferences that informed the study design, recruitment as well as the design and delivery of the intervention.

**Contributors** DS, MB and WID contributed to the inception of the study. FH, MB, LC, CD, WID, DL, VP, TP-H, ES and DS contributed to the study design and planning of the protocol. FH, DS, MB, WID, CD and VP contributed to the design of the intervention. FH and DS drafted the manuscript and FH, MB, LC, CD, WID, DL, VP, TP-H, ES and DS critically revised and approved the manuscript.

**Funding** We gratefully acknowledge the National Institute for Health Research–Research for Patient Benefit Programme for providing funds to support this project (grant number PB-PG-0215-36091). This paper reports independent research funded by the National Institute for Health Research (Research for Patient Benefit Programme, PB-PG-0215-36091). The views expressed in the publication are those of the authors and not necessarily those of the NIHR or the Department of Health and Social Care.

**Competing interests** None declared.

**Patient consent for publication** Not required.

**Ethics approval** Favourable ethical approval was obtained from the Health Research Authority and London—Fulham Research Ethics Committee and (17/LO/2043).

**Provenance and peer review** Not commissioned; externally peer reviewed.

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
