## [Reviewer comments · BMJ Open]

ARTICLE DETAILS

TITLE (PROVISIONAL)	Protocol for a feasibility randomised controlled trial of a musculoskeletal exercise intervention versus usual care for children with haemophilia
AUTHORS	Hashem, Ferhana; Bladen, Melanie; Carroll, Liz; Dodd, Charlene; Drechsler, Wendy; Lowery, David; Patel, Vishal; Pellatt-Higgins, T; Saloniki, Eirini; Stephensen, David

VERSION 1 - REVIEW

REVIEWER	Rubén Cuesta-Barriuso Department of Physiotherapy, European University of Madrid, Spain. Real Fundación Victoria Eugenia, Madrid, Spain. Fishemo CEE. Spanish Federation of Hemophilia, Madrid, Spain.
REVIEW RETURNED	25-Feb-2019

GENERAL COMMENTS	This is an extensive article where, mainly, a review of the methodology to be used in a randomized clinical study is made. However, there are issues that should be considered when proposing a clinical study and that would give greater meaning to this study: - The calculation of the sample size can be done in multiple ways (prevalence, partners in an NMO, national registry of patients, etc.), so it would not make sense to be an objective / study variable.- Forensic medicine stipulates professional and socio-health costs according to pathologies and damages. The use of these standardized tables would be more correct, as it is not clear how this variable will be measured.- When talking about usual care what do the authors refer to? Are there standardized treatments of muscle strength (usual care) for children of 6-11 years without apparent joint damage?- In the introduction it is said that there is growing evidence of the benefits of muscle strength training. However, it is indicated how the Cochrane review throws away publications because of their lack of methodological quality. It is an adventure to talk about evidence when there are few and bad clinical studies, and even more so in children with hemophilia.- The inclusion criteria are not very clear: Would children with hemophilia A, B or von Willebrand disease be included ?; severe, moderate or mild hemophilia? In prophylactic treatment or on demand? With or without joint damage in some joint? If there is joint damage, to what degree is it established that there is joint damage already?
--

	 - The intervention would be the same for children of 6 and 11 years? The biomechanical characteristics of stability and muscular strength are not the same nor have they developed in the same way: how is the difference in this range to be adjusted? - In the intervention it is said that the progressive intensity will be increased: under what parameters will the intensity be increased? - The blinding of the evaluator in simple and is very well referenced in the methodological quality guides: at no time will be aware of the assignment to the groups and will not ask (nor be asked) about any data that is not a consequence of the evaluation of the dependent variables. - Within the primary outcomes measures, they comment that they will value the cost of factor concentrates. They should indicate which of the three possible factor consumption data they are going to use (what is prescribed by the doctor, what is delivered to the hospital and what the patient actually administers) and how they will reject the factor expense that is really a consequence of the intervention (for bad and for good). This question is not clear, considering that we are talking about children with hemophilia with little / no joint damage. - Within the secondary outcomes measures, the first four variables have no meaning in the design of an RCT since it is assumed that they must be logical for this type of design, regardless of whether they are related to the objectives of the study. - It should be indicated which joints are going to be evaluated since in the introduction only the prevalence of ankle arthropathy is indicated, and as we do not know which regions are going to work, we can not know which joints will be evaluated. The ultrasound study and with the HJHS scale will be done on knees, elbows and ankles? - Despite all these comments, the big problem of the article is that it repeats many data and arguments that can be found in any guide of methodological quality of a randomized clinical trial but the fundamental information of a RCT on strength in hemophilia omits the intervention to be performed.
--	---

REVIEWER	Dr Janjaap van der Net Centre for Child Development, Exercise Center and Physical literacy University Medical Center and Children's Hospital Utrecht, The Netherlands.
REVIEW RETURNED	18-Mar-2019

GENERAL COMMENTS	 - It is a concern that the theoretical basis of this study is based on a phenotype (and epidemiology) that is more than 10 yrs old. - The notion that despite prophylaxis, boys still bleed 1-2 times per year, is based on reports that are resp. 13, 12 and 10 yrs old (ref 1, 7,8). One might question if this is still valid? The biomechanics en muscle dynamics studies are of more recent date and might be much more supportive to this study (ref 13,14). - The "growing evidence of the benefits of incorporating muscle strengthening exercise in the management of arthropathy" is not substantiated with references. - The rationale for this ... (p 4; line 57-p 5; line 5) is primarily opinion based, not evidence based. - The primary outcome measures are "self reported" measures, whereas the majority of the secondary outcome measures are more objective measures/scores. Especially the report of the number of exercises completed might be prone to over or under
--

	report in developmental psychology it is a well known observation that teenagers might be poor reporters on their activities).
--	--

VERSION 1 – AUTHOR RESPONSE

Reviewer: 1

1.1 The calculation of the sample size can be done in multiple ways (prevalence, partners in an NMO, national registry of patients, etc.), so it would not make sense to be an objective / study variable.

Estimates of effect size of the key variables, muscle strength and joint motion in children with haemophilia that we hypothesise to detect in this study are not currently available from national patient registries and prevalence data and as such cannot be used to estimate the sample size required to test out hypothesis. Furthermore, there is currently no published data testing the efficacy of a muscle strengthening intervention in children with haemophilia in a randomised trial to accurately estimate sample size. We have based an estimate of effect size and subsequent sample size likely to be required on differences between haemophilia boys and healthy peers from our previous observation cross-sectional studies of muscle strength and joint biomechanics. Data from the current feasibility study will provide a more informed estimate of the effect size and the sample size required to test the efficacy of the intervention in an appropriately powered study. It is our opinion that this remains a key objective of the feasibility study and thus no changes have been made to the manuscript regarding this point.

1.2 Forensic medicine stipulates professional and socio-health costs according to pathologies and damages. The use of these standardized tables would be more correct, as it is not clear how this variable will be measured.

The study protocol submitted with the manuscript as a supplementary file contains additional detail regarding how we will measure costs relevant to the study. Costs relevant to the study will be divided into healthcare sector and patient and family costs. Healthcare costs (using National Health Service reference costs where possible) will include; training and delivering the intervention, equipment, overhead costs, visits to hospital/ haemophilia clinic, general practitioner, other healthcare services, investigations, hospitalisation, treatment, coagulation clotting factor concentrate and medication. Patient and family costs will include travel and time off work. Feasibility of collecting these data will be explored along with the resource consequences. This detail has now been added to the manuscript (Page 10, Line 4-10).

1.3 When talking about usual care what do the authors refer to? Are there standardized treatments of muscle strength (usual care) for children of 6-11 years without apparent joint damage?

Physiotherapists in the UK are autonomous clinicians who use clinical reasoning to provide a range of interventions as part of usual care. There are currently no standardised treatment recommendations

for children without apparent joint damage. This sentence has been added (Page 7, Line 34). The original manuscript outlines what “usual care” entails.

1.4 In the introduction it is said that there is growing evidence of the benefits of muscle strength training. However, it is indicated how the Cochrane review throws away publications because of their lack of methodological quality. It is an adventure to talk about evidence when there are few and bad clinical studies, and even more so in children with hemophilia.

We agree that the evidence for muscle strengthening is of low quality, hence our use of the term “growing evidence” to indicate emergence of evidence rather than using the term “strong evidence”. To avoid misinterpretation we have amended “growing evidence to “emerging evidence”. We have already clarified this statement in the original manuscript in reference to the Cochrane review referred to;” a recent Cochrane Review evaluating the safety and effectiveness of exercise for people with haemophilia reported four randomly controlled studies of an exercise intervention in children with the condition and concluded the studies were of low or very low quality, due to small sample sizes and potential bias. Furthermore, no paediatric study compared a muscle strengthening intervention to a control group or intervention without muscle strengthening exercises. Several studies compared two multi-component exercise interventions that included broad muscle strengthening as well as treadmill walking and cycle ergometry and it is not possible to determine the effect of muscle strengthening on muscle strength or its effect on physical function.”

1.5 The inclusion criteria are not very clear: Would children with hemophilia A, B or von Willebrand disease be included ?; severe, moderate or mild hemophilia? In prophylactic treatment or on demand? With or without joint damage in some joint? If there is joint damage, to what degree is it established that there is joint damage already?

We have modified the inclusion criteria to state that children with haemophilia A or B on prophylactic treatment with coagulation factor concentrates (Page 5, Line 30-31). We had already stated in the original manuscript that children with severe or moderate haemophilia with or without inhibitors would be included.

1.6 The intervention would be the same for children of 6 and 11 years? The biomechanical characteristics of stability and muscular strength are not the same nor have they developed in the same way: how is the difference in this range to be adjusted?

It is well established that kinematic gait characteristics have matured by the age of 6 and that muscle strength increases linearly with age and body mass until puberty. In our analysis we will normalise walking variables to leg length and muscle strength to body mass. This information is now included in the manuscript (Page 10, Line 29-30). We have already stated in the original manuscript that, “change from baseline data will be compared between groups at all study time points using analysis of covariance with baseline as a covariate and a factor for centre. Estimates of differences between treatment arms in the changes from baseline (adjusted for baseline), and 75% and 95% confidence intervals will be calculated”.

1.7 In the intervention it is said that the progressive intensity will be increased: under what parameters will the intensity be increased?

The exercise programme is progressed in terms of intensity (increase in repetitions) and load (increase in muscle control to perform the exercise). A copy of the exercise programme is now included as Figure 2.

1.8 The blinding of the evaluator is simple and is very well referenced in the methodological quality guides: at no time will be aware of the assignment to the groups and will not ask (nor be asked) about any data that is not a consequence of the evaluation of the dependent variables.

We agree that blinding of the evaluator is possible in this study and we have stated in the original manuscript we will attempt a single blind approach, in which the participant and physiotherapists delivering the intervention will be encouraged to withhold their group allocation from the assessors collecting data. We will use a range of strategies, including: instruction in the participant literature; intervention protocol and with frequent verbal prompts. In order to assess the efficacy of blinding, we will ask assessors to record which group they think participants have been allocated to at the end of each assessment. This will be presented as a binary forced choice. There are no circumstances under which it would be necessary to un-blind assessors.

1.9 Within the primary outcomes measures, they comment that they will value the cost of factor concentrates. They should indicate which of the three possible factor consumption data they are going to use (what is prescribed by the doctor, what is delivered to the hospital and what the patient actually administers) and how they will reject the factor expense that is really a consequence of the intervention (for bad and for good). This question is not clear, considering that we are talking about children with hemophilia with little / no joint damage.

Factor consumption data will be recorded as “what the patient actually administers” which is recorded on a web-based system as part of routine care. We have clarified this in the revised manuscript (Page 8, Line 12). As stated in the original manuscript data including annualised bleeding rates (ABR) and coagulation factor usage will be analysed as “change from baseline data will be compared between groups at all study time points using analysis of covariance with baseline as a covariate. Estimates of differences between treatment arms in the changes from baseline (adjusted for baseline), and 75% and 95% confidence intervals will be calculated”. Annualised bleeding rates (ABR) and coagulation factor usage in the 12 weeks before commencing the intervention will be used as baseline data. Any change will be assumed to be related to the intervention.

1.10 Within the secondary outcomes measures, the first four variables have no meaning in the design of an RCT since it is assumed that they must be logical for this type of design, regardless of whether they are related to the objectives of the study.

The aim of this study is to assess feasibility not efficacy, hence the secondary outcomes of recruitment rate, willingness to be randomised, participant acceptability and training required to deliver the intervention are important in terms of whether it is possible to overcome the challenges of undertaking a fully powered study.

1.11 It should be indicated which joints are going to be evaluated since in the introduction only the prevalence of ankle arthropathy is indicated, and as we do not know which regions are going to work, we can not know which joints will be evaluated. The ultrasound study and with the HJHS scale will be done on knees, elbows and ankles?

The full HJHS and ultrasound score will be completed on right and left ankle, knee and elbow joints. This information is already included for ultrasound imaging in the original manuscript and has now been included for the HJHS (Page 9, Line 2-3).

1.12 Despite all these comments, the big problem of the article is that it repeats many data and arguments that can be found in any guide of methodological quality of a randomized clinical trial but the fundamental information of a RCT on strength in hemophilia omits the intervention to be performed.

A copy of the exercise programme is now included as Figure 2.

Reviewer: 2

2.1 It is a concern that the theoretical basis of this study is based on a phenotype (and epidemiology) that is more than 10 yrs old. The notion that despite prophylaxis, boys still bleed 1-2 times per year, is based on reports that are resp. 13, 12 and 10 yrs old (ref 1, 7,8). One might question if this is still valid? The biomechanics en muscle dynamics studies are of more recent date and might be much more supportive to this study (ref 13,14).

We agree that one might expect that recent advances in medical treatment would lead to reduced bleeding episodes and musculoskeletal impairment. A recent review of patient registry data of 199 children in the United Kingdom from 2015 reported a median annual bleeding rate of one bleed per year (IQR: 0.0-5.0) in those aged 0-11 years with severe haemophilia receiving prophylaxis. Annual bleeding rates in a further 55 children aged 0-11 years with moderate haemophilia were 3.0 (1.0-7.0) in those reporting prophylaxis and 11.0 (4.8-20.3) in those treated on-demand. The reference has been added to the introduction (Page 4, Line 11) and Reference List: Scott MJ, Xiang H, Hart DP, Palmer B, Collins PW, Stephensen D, Sima CS, Hay CRM. Treatment regimens and outcomes in severe and moderate haemophilia A in the UK: The Thunder Study. *Haemophilia*, 2019; 25(2):205-212.

Together with biomechanical and muscle function data we believe the rationale for the study remains current.

2.3 The "growing evidence of the benefits of incorporating muscle strengthening exercise in the management of arthropathy" is not substantiated with references.

The relevant statement has now been referenced (Page 4, Line32)

2.4 The rationale for this ...(p 4; line 57-p 5; line 5) is primarily opinion based, not evidence based.

We agree that there is a lack of evidence to support these assumptions and have added additional text for clarity (Page 4, Line 32-33).

2.5 The primary outcome measures are "self reported" measures, whereas the majority of the secondary outcome measures are more objective measures/scores. Especially the report of the number of exercises completed might be prone to over or under report in developmental psychology it is a well known observation that teenagers might be poor reporters on their activities).

We agree self-reported measures like the number of exercises completed might be prone to over or under reporting in children and is a limitation of this data. We will employ strategies to encourage accurate reporting of data. The study will only recruit children aged 6-11 years and the exercises will be supervised once by the physiotherapist and any further times by the child's parent. All self recorded data will be recorded in participant diaries by the children's parents. The physiotherapist delivering the intervention will visit the participant's home once per week and contact the participant's parent once further each week to encourage accuracy of self-reported data. The physiotherapist will collect the diary every 2 weeks to ensure it is being completed.

VERSION 2 – REVIEW

REVIEWER	Janjaap van der Net Center for Childdevelopment, exercise and physical literacy, university children's hospital Utrecht, The Netherlands
REVIEW RETURNED	01-May-2019

GENERAL COMMENTS	The authors are succeeded to implement all reviewers suggestions
--